# Mechano-redox control of integrin de-adhesion

Freda Passam[1†], Joyce Chiu[2,3†], Lining Ju[4], Aster Pijning[2], Zeenat Jahan[1], Ronit Mor-Cohen[5], Adva Yeheskel[6], Katra Kolšek[7,8], Lena Thärichen[7,8], Camilo Aponte-Santamaría[7,9], Frauke Gräter[7,8], Philip J Hogg[2,3]*

[1]St George Clinical School, Kogarah, Australia; [2]The Centenary Institute, Newtown, Australia; [3]National Health and Medical Research Council Clinical Trials Centre, University of Sydney, Sydney, Australia; [4]Heart Research Institute and Charles Perkins Centre, University of Sydney, Sydney, Australia; [5]The Amalia Biron Research Institute of Thrombosis and Hemostasis, Sheba Medical Center, Tel Hashomer and Sackler Faculty of Medicine, Tel Aviv University, Tel Aviv, Israel; [6]The Bioinformatics Unit, George S. Wise Faculty of Life Science, Tel Aviv University, Tel Aviv, Israel; [7]Heidelberg Institute of Theoretical Studies, Heidelberg, Germany; [8]Interdisciplinary Center for Scientific Computing, Heidelberg University, Heidelberg, Germany; [9]Max Planck Tandem Group in Computational Biophysics, University of Los Andes, Bogotá, Colombia

**Abstract** How proteins harness mechanical force to control function is a significant biological question. Here we describe a human cell surface receptor that couples ligand binding and force to trigger a chemical event which controls the adhesive properties of the receptor. Our studies of the secreted platelet oxidoreductase, ERp5, have revealed that it mediates release of fibrinogen from activated platelet αIIbβ3 integrin. Protein chemical studies show that ligand binding to extended αIIbβ3 integrin renders the βI-domain Cys177-Cys184 disulfide bond cleavable by ERp5. Fluid shear and force spectroscopy assays indicate that disulfide cleavage is enhanced by mechanical force. Cell adhesion assays and molecular dynamics simulations demonstrate that cleavage of the disulfide induces long-range allosteric effects within the βI-domain, mainly affecting the metal-binding sites, that results in release of fibrinogen. This coupling of ligand binding, force and redox events to control cell adhesion may be employed to regulate other protein-protein interactions.
DOI: https://doi.org/10.7554/eLife.34843.001

*For correspondence:
phil.hogg@sydney.edu.au

[†]These authors contributed equally to this work

Competing interests: The authors declare that no competing interests exist.

## Introduction

Protein function is controlled by a variety of chemical modifications to amino acid side chains, cleavage or isomerization of peptide bonds and cleavage or formation of disulfide bonds (*Butera et al., 2014a*; *Cook and Hogg, 2013*). These chemical reactions are usually facilitated by enzymes and their cofactors. Mechanical force is another factor that is increasingly being recognized to control protein chemical reactions (*Garcia-Manyes and Beedle, 2017*; *Cross, 2016*). Mechanical force can markedly reduce the reaction energy barrier. Reactions that are too slow become relevant on a biological time scale and others would not occur at all without the input of force. Low forces can trigger bond rotation and rupture of hydrogen bonds, while high forces can break or form covalent bonds. A number of cell surface receptors have been shown to be regulated by mechanical forces (*Chen et al., 2017*), including the integrins.

Vertebrates express 24 different integrins that comprise one of 18 different α-subunits and one of 8 different β-subunits (*Hynes, 2002*). Integrin-mediated adhesion and signalling events regulate

**eLife digest** Many proteins embedded in a cell's surface allow the cell to interact with its surroundings. Integrins are a group of cell surface proteins that have many uses in different cells. Integrins become activated when they come into contact with other specific proteins, which like other molecules that bind to proteins are referred to collectively as "ligands". Much research has focused on how ligands become attached to integrins and how this activates these cell surface proteins. Yet how integrins release ligands and become inactive has not been studied before.

One type of integrin, called αIIbβ3, is involved in blood clotting. Found on the surface of blood platelets – the fragments of cells in the blood that play a central role in clotting, this integrin binds to a ligand called fibrinogen. Fibrinogen links platelets together to form clots by building bridges between integrins. Passam, Chiu et al. have now studied platelets from donated human blood to understand how the integrin αIIbβ3 disengages from fibrinogen.

The investigation showed that an enzyme called ERp5 aids the release of fibrinogen from the integrin. ERp5 can be released by blood vessel walls and by activated platelets. The experiments revealed that ERp5 breaks a chemical link, called a disulfide bond, in the integrin, but only when the protein is already bound to its ligand. Breaking the disulfide bond (a chemical process known as reduction) changes the integrin's structure so that it lets go of fibrinogen. Moreover, when physical forces such as blood flow put the integrin under strain, the ERp5 enzyme becomes more effective.

These findings show how ligand binding and mechanical force work together to control the breaking of a chemical bond in a human integrin. This chemical event then in turn controls the release of the integrin's ligand. It is possible that other protein-protein interactions may involve similar mechanisms, but this remains to be explored.

Lastly, Passam, Chiu et al. suggest that the release of fibrinogen might help to limit the growth of blood clots so they do not block the blood vessels. Further studies should test this hypothesis. Inappropriate clotting can have severe health effects including heart attacks and strokes. As such, this investigation may hint at a more subtle way to regulate clotting through integrin αIIbβ3, such as boosting fibrinogen release to see if it helps slow or reduce clotting without stopping it altogether.
DOI: https://doi.org/10.7554/eLife.34843.002

virtually all cell growth and differentiation, while dysregulation of integrins are involved in the pathogenesis of cancer, auto-immune conditions and vascular thrombosis. The integrin heterodimers recognize overlapping but distinct sets of ligands on other cells, extracellular matrix or on pathogens. The integrins are type one trans-membrane proteins that consist of large extracellular segments characterized by various domains and small transmembrane and cytoplasmic segments.

Most integrins exist on the cell surface in an inactive state that does not bind ligand or signal. Integrins are activated by intracellular stimuli such as talin binding through a process termed inside-out signalling, and by ligand occupancy that transduces extracellular signals into the cytoplasm through outside-in signalling (*Tadokoro et al., 2003*; *Luo et al., 2007*). Affinity for ligands is mostly controlled by global and local conformational rearrangements of the integrin ectodomains. The extracellular segments exist in at least three conformational states: the bent conformation with closed headpiece (low affinity for ligand), the extended conformation with closed headpiece (intermediate affinity) and the extended conformation with open headpiece (high-affinity) (*Zhu et al., 2013*). While much is known about the structure and function of resting and activated integrins, little is known about how integrins disengage from their ligands.

For instance, when cells migrate, integrins adhere at the leading edge and de-adhere at the trailing edge. Force has been suggested as one of the mechanisms to explain both the engagement and disengagement of ligand (*Zhu et al., 2008*). It was proposed that binding of the integrin β subunit cytoplasmic domain to actin filaments results in lateral translocation of the integrin heterodimer on the cell surface that causes integrin extension. Engagement of immobilised extracellular ligand greatly increases this lateral force that favours the high-affinity, open headpiece conformation. Disassembly of the actin cytoskeleton and dissociation of the β subunit cytoplasmic domain removes the lateral force that results in the closed headpiece conformation and ligand disengagement. In support of this model, cells become highly elongated and stop migrating when integrins are locked in the

high-affinity conformation or when actin disassembly is blocked in the uropod (*Smith et al., 2007*). Here we report a chemical modification of an activated integrin that results in disengagement of ligand.

Platelet clumping at sites of blood vessel injury is mediated by cross-linking of platelet αIIbβ3 integrin by the bivalent ligand, fibrinogen. This integrin is critical for thrombus formation and is the target of successful anti-thrombotic agents in routine clinical use for acute coronary syndrome. ERp5 is a protein disulfide isomerase (PDI) family member oxidoreductase released from platelets upon activation (*Jordan et al., 2005*), and from platelets and endothelial cells at sites of thrombosis in mice (*Passam et al., 2015*). Human platelets contain about 13,300 molecules of ERp5 per platelet (*Burkhart et al., 2012*), while mouse platelets contain an estimated 60,000 molecules (*Zeiler et al., 2014*). Secreted ERp5 binds to the β3 subunit of platelet surface αIIbβ3 integrin (*Jordan et al., 2005*; *Passam et al., 2015*). Systemic inhibition of ERp5 with function-blocking antibodies inhibits thrombosis in mice, which implies an essential role for secreted ERp5 in this biology (*Passam et al., 2015*).

Our studies indicate that ERp5 mediates de-adhesion of activated platelet αIIbβ3 integrin from fibrinogen. Ligand binding to extended αIIbβ3 integrin triggers cleavage of the βI-domain Cys177-Cys184 disulfide by ERp5, which is enhanced by mechanical force. Cleavage of the disulfide results in release of fibrinogen due to allosteric effects at the metal-ion-dependent adhesion (MIDAS) site.

## Results

### ERp5 triggers fibrinogen dissociation from αIIbβ3 integrin

Immunoblotting of human platelet lysate and releasate indicates that approximately half of the platelet ERp5 molecules are released into the supernatant upon activation (*Figure 1—figure supplement 1*). ERp5 binds to β3 integrin with a dissociation constant in the low micromolar range (*Passam et al., 2015*), and an anti-ERp5 antibody has been reported to inhibit fibrinogen binding to activated platelets and platelet aggregation in vitro (*Jordan et al., 2005*). These findings suggested that ERp5 directly regulates platelet αIIbβ3 integrin function, although the mechanism remains elusive. We examined the effect of soluble ERp5 on platelet αIIbβ3 integrin activation and fibrinogen binding.

Incubation of washed platelets with ERp5 did not trigger αIIbβ3 integrin activation, and ERp5 had no effect on integrin activation by ADP (*Figure 1A*). Integrin activation was measured by binding of PAC-1, an antibody that recognizes the fully activated integrin with an open headpiece (*Shattil et al., 1985*; *Luo et al., 2003*). This result indicated that ERp5 is not directly involved in integrin activation, so we explored a role for ERp5 in post-activation events. Effect of soluble ERp5 on the kinetics of adhesion of washed platelets to fibrinogen as a function of fluid shear force was examined (*Figure 1B*). There was a biphasic effect of ERp5 on platelet binding to fibrinogen at two wall shear stress (*Figure 1C*). At 10 dyn/cm$^2$ (1000 s$^{-1}$) platelet adhesion in the first 4 min of flow was unaffected by ERp5 and reduced with time thereafter. The negative effect of ERp5 on platelet adhesion was more pronounced at 30 dyn/cm$^2$ (3000 s$^{-1}$). Platelet adhesion in the first 1 min of flow was unaffected by ERp5 and reduced significantly with time thereafter. This finding suggested that ERp5 was triggering dissociation of fibrinogen from activated platelet αIIbβ3 integrin. To better understand this mechanosensitive phenomenon, the effect of ERp5 on binding of fibrinogen to αIIbβ3 integrin was characterized using a force spectroscopy technique - biomembrane force probe (BFP) (*Ju et al., 2016*, *Ju et al., 2013*).

The BFP brings an αIIbβ3 integrin coated bead into contact with a fibrinogen bearing force probe (*Figure 2A*). Upon target retraction, it detects the αIIbβ3 bond from the pico-force signal measured (*Figure 2B*, red), while a zero force indicates a no-bond event (*Figure 2B*, black). The intermittent 'touch and retract' cycles mimic platelet translocation behavior under shear (*Ju et al., 2013*; *Yago et al., 2008*). The adhesion frequencies (the number of 'bond' touches (*Figure 2F*, red) divided by the number of total touches) reflects the binding affinity at zero tensile force. Unexpectedly, we found that soluble ERp5 had no significant effect on the αIIbβ3–fibrinogen adhesion frequencies (*Figure 2C*). To investigate the force effect, we measured αIIbβ3 integrin–fibrinogen bond lifetimes at multiple clamped forces in the absence or presence of ERp5. This interaction is characterized by slip-bond behaviors in which force accelerates bond dissociation, consistent with the

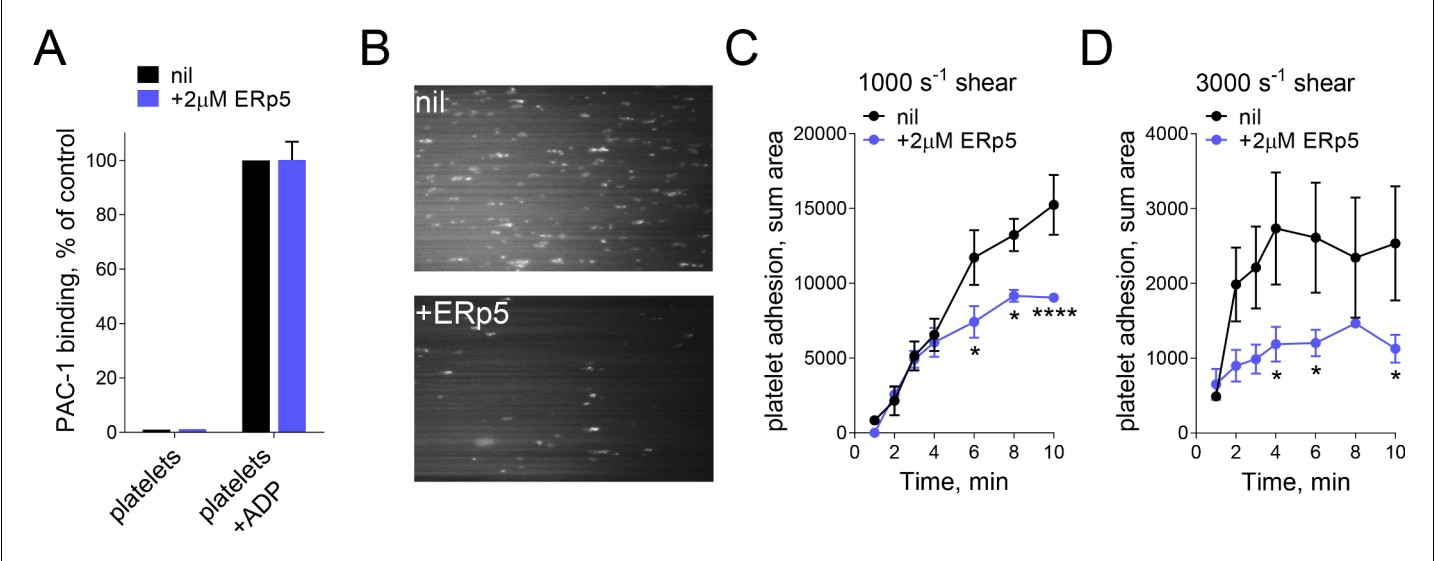

**Figure 1.** ERp5 triggers αIIbβ3 integrin de-adhesion from fibrinogen. (A) ERp5 does not induce αIIbβ3 integrin activation on platelets, or influence integrin activation by ADP. The activated αIIbβ3 is measured by PAC-1 antibody binding. The data points and errors (1 SD) are from measurements of four different healthy donor platelets. (B–D) Platelet adhesion to fibrinogen under flow is impaired by ERp5. Part B are representative images of platelet adhesion to immobilized fibrinogen in the absence of presence of 2 μM ERp5. Parts C and D are kinetics of platelet adhesion to fibrinogen in the absence or presence of 2 μM ERp5 at fluid shear rates of 1000 s$^{-1}$ or 3000 s$^{-1}$, respectively. The data points and errors (1 SD) are from three measurements each from three different healthy donor platelets. *p<0.05, ****p<0.001; assessed by unpaired, two-tailed Student's t-test.
DOI: https://doi.org/10.7554/eLife.34843.003

The following figure supplement is available for figure 1:

**Figure supplement 1.** ERp5 level in human platelets and platelet releasate.
DOI: https://doi.org/10.7554/eLife.34843.004

previous optical tweezer study (*Litvinov et al., 2011*). Surprisingly, at the low force regime of 5–15 pN, ERp5 displayed a similarly non-significant effect on the αIIbβ3–fibrinogen bond lifetimes as the adhesion frequency assay, whereas as force goes beyond 15 pN, it greatly enhanced αIIbβ3–fibrinogen dissociation with reduced bond lifetimes (*Figure 2D*). In accordance with the perfusion experiments, these findings indicate that ERp5 has a force-dependent de-adhesive effect on the αIIbβ3–fibrinogen interaction.

## ERp5 cleaves the βI-domain Cys177-Cys184 disulfide bond

ERp5 is an oxidoreductase that can cleave, form or potentially rearrange disulfide bonds in protein substrates. From crystal structures of the complete ectodomain of αIIbβ3 integrin (*Zhu et al., 2008*), 28 disulfide bonds have been defined in the β3 subunit. No unpaired cysteines within a few Angstoms of each other were identified, which implied that all possible disulfide bonds in the subunit are intact. We hypothesized that the functional effect of ERp5 on fibrinogen binding was a result of its cleavage of one or more of the 28 β3 disulfide bonds. This was tested by measuring the presence of unpaired cysteine thiols in platelet surface β3 before and after activation by ADP. Platelet activation resulted in increased labelling of β3 by a thiol-specific probe (*Figure 3A*), indicating that one or more disulfide bonds were cleaved in the subunit upon platelet activation.

To identity the β3 disulfide(s) cleaved by ERp5, it was critical that we accurately quantify the redox state of the subunits disulfide bonds. This was achieved using a differential cysteine alkylation and mass spectrometry technique (*Pasquarello et al., 2004*; *Bekendam et al., 2016*). Briefly, reduced disulfide bond cysteines in purified platelet β3 integrin were alkylated with 2-iodo-N-phenylacetamide (IPA), and the oxidized disulfide bond cysteines with a stable carbon-13 isotope of IPA following reduction with dithiothreitol (*Figure 3B*). Sixty-eight cysteine containing peptides (*Supplementary file 1*) reporting on 24 of the 28 β3 integrin disulfides were resolved by mass spectrometry and quantified (*Figure 3—figure supplement 1*). The four disulfides we were unable to map are the Cys528-Cys542 and Cys536-Cys547 bonds of the EGF-3 domain, Cys575-Cys586 of the

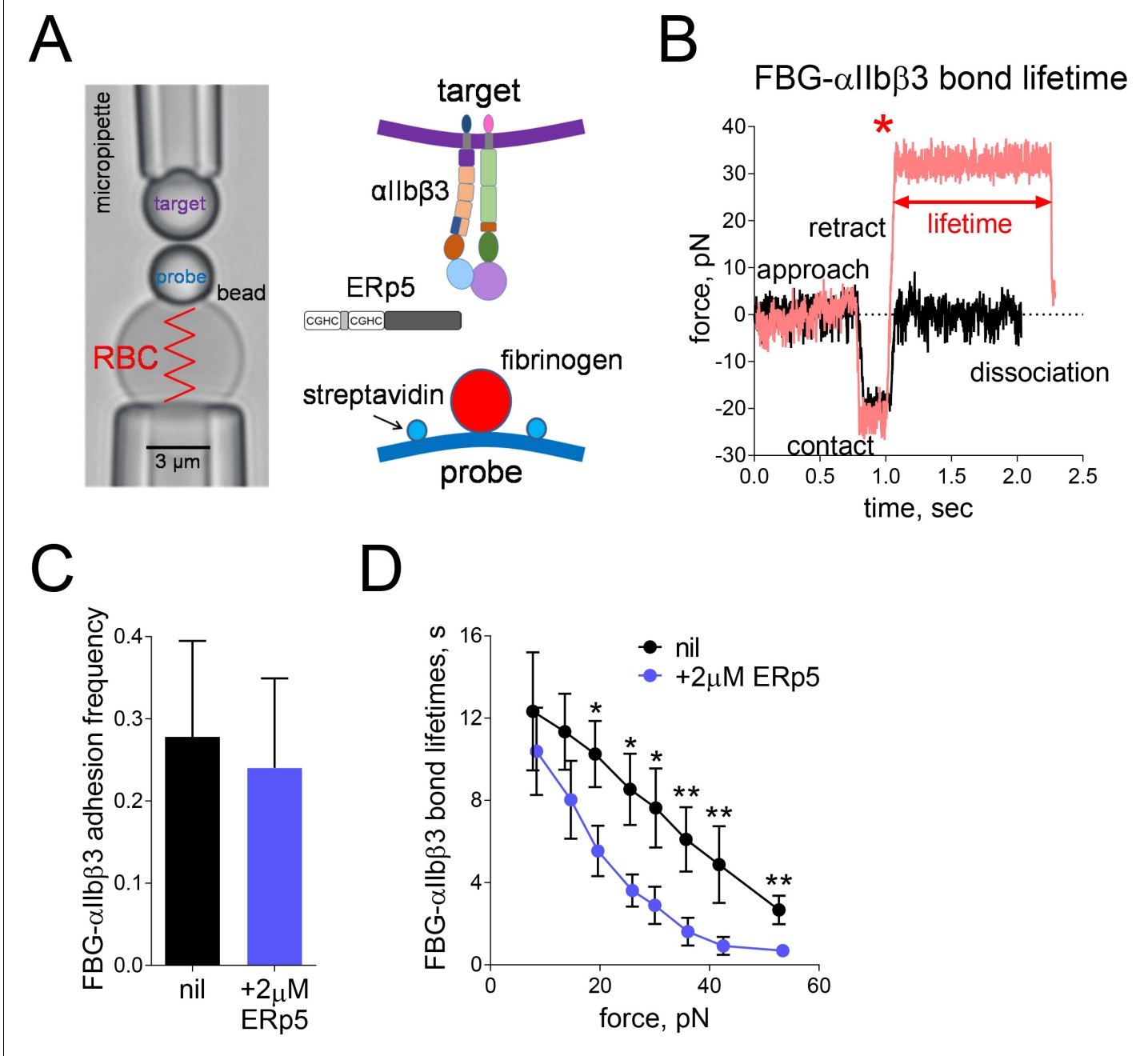

**Figure 2.** Mechanical force accelerates ERp5-induced fibrinogen de-adhesion. (A) Biomembrane-Force-Probe (BFP) detection of αIIbβ3–fibrinogen (FBG) binding in a purified system. The probe bead was coated with fibrinogen and streptavidin (SA) for attachment of the bead to biotinylated red blood cell (RBC). Fibrinogen is the focus for interaction with αIIbβ3 on the target bead. The effect of soluble ERp5 on this interaction was measured. (B) Force versus time traces from two representative test cycles. The target bead was driven to approach and contact the probe bead, then retracted. In a 'no bond' event (black), the cycle ended after the probe–target separation. In a 'bond' event (red), the target was clamped (marked by *) at a preset force until dissociation. Lifetime is measured from the point when the clamped force (i.e. 30 pN) was reached to the point when the bond dissociated, signified by a force drop to zero. (C) Soluble ERp5 has no significant effect on αIIbβ3–fibrinogen adhesion frequency. The adhesion frequencies represent mean ± SEM of n = 3 independent experiments in the absence or presence of 2 μM ERp5. For each experiment, five random probe–target pairs with 50 touches were analyzed and averaged. (D) Lifetime of fibrinogen–αIIbβ3 integrin bonds vs. clamp force in the absence or presence of 2 μM ERp5. Results represent mean ±SEM of 10–30 measurements at each force bin. *p<0.05, **p<0.01; assessed by unpaired, two-tailed Student's t-test.
DOI: https://doi.org/10.7554/eLife.34843.005

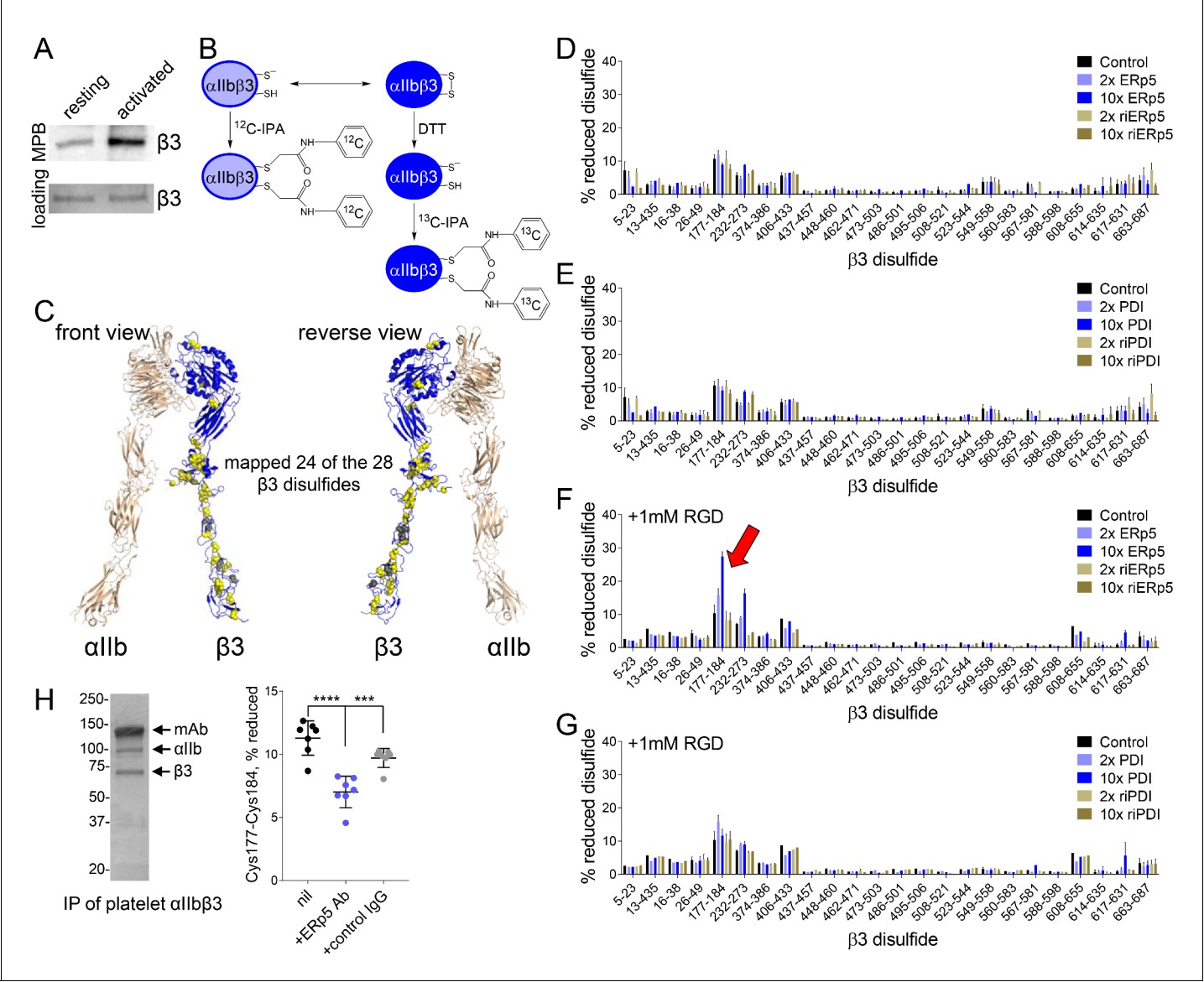

**Figure 3.** ERp5 cleaves the βI-domain Cys177-Cys184 disulfide bond. (A) One or more disulfide bonds are cleaved in platelet β3 integrin upon platelet activation. Washed platelets were untreated or activated with ADP and unpaired cysteine thiols in platelet surface proteins labelled with the biotin-linked maleimide, MPB. The β3 integrin was immunoprecipitated from the platelet lysate and blotted for MPB. (B) Differential cysteine alkylation and mass spectrometry method of measuring the redox state of the β3 integrin cysteines. Unpaired cysteine thiols in purified β3 integrin are alkylated with $^{12}C$-IPA and the disulfide bonded cysteine thiols with $^{13}C$-IPA following reduction with DTT. Sixty-eight peptides (*Supplementary file 1*) encompassing cysteines representing 24 of the 28 β3 integrin disulfide bonds were analyzed. (C) Positions of the β3 integrin disulfide bonds (yellow and grey spheres) in a modelled open structure of the complete αIIbβ3 integrin ectodomain (blue ribbon). We were able to map 24 of the 28 β3 disulfide bonds. The mapped disulfides are in yellow and the unmapped bonds are in grey. (D–E) Redox state of 24 of the 28 β3 integrin disulfide bonds in the absence or presence of 2- or 10-fold molar excess of ERp5 or redox inactive ERp5 (D), or PDI or redox inactive PDI (E). (F–G) Redox state of 24 of the 28 β3 integrin disulfide bonds in the absence or presence of 2- or 10-fold molar excess of ERp5 or redox inactive ERp5 (D), or PDI or redox inactive PDI (E), and 1 mM RGD peptide. The bars and errors (1 SD) are for 5–15 measurements from three different integrin preparations. The β3 Cys177-Cys184 disulfide bond (indicated by red arrow) is significantly cleaved by 10-fold molar excess of ERp5 (p<0.05) (H) The βI-domain Cys177-Cys184 disulfide bond is cleaved on the platelet surface by platelet ERp5. Washed platelets were incubated with function-blocking anti-ERp5 antibodies or isotype control antibodies and the redox state of the βI-domain disulfide determined in the integrin immunoprecipitated from lysate. The bars and errors (1 SD) are from three different platelet preparations from healthy donors. ***p<0.005; ****p<0.001; assessed by unpaired, two-tailed Student's t-test.

DOI: https://doi.org/10.7554/eLife.34843.006

The following figure supplements are available for figure 3:

**Figure supplement 1.** Differential cysteine alkylation and mass spectrometry analysis of the β3 Cys177-Cys184 disulfide bond.

*Figure 3 continued on next page*

*Figure 3 continued*

DOI: https://doi.org/10.7554/eLife.34843.007

**Figure supplement 2.** Redox state of the β3 Cys177-Cys184 disulfide bond in the absence or presence of 10-fold molar excess of ERp5 and 1 mM RGD or control RGE peptide.

DOI: https://doi.org/10.7554/eLife.34843.008

EGF-4 domain and Cys601-Cys604 of the Ankle domain (*Figure 3C*). Purified platelet αIIbβ3 integrin was incubated with 2- or 10-fold molar excess of ERp5 or PDI and the redox state of the disulfides quantified. PDI was used to test the substrate specificity of ERp5. PDI is the archetypal member of the PDI family of oxidoreductases, which includes ERp5, and is also secreted by platelets and endothelial cells at sites of thrombosis in mice and binds to surface β3 integrin (*Cho et al., 2008*; *Cho et al., 2012*). Reactions were also performed with redox-inactive ERp5 and PDI to test the redox dependence of their action. Redox-inactive ERp5 and PDI were produced by replacing the active-site cysteines with alanines.

In accordance with crystal structures of the integrin ectodomain, all 24 disulfide bonds in untreated β3 integrin were >90% oxidized, with one exception (*Figure 3D and E*). Approximately 10% of the βI-domain Cys177-Cys184 bond was reduced in the β3 preparations. Incubation of the integrin with ERp5 or PDI did not significantly change the redox sate of any of the 24 disulfide bonds (*Figure 3D and E*). Our fibrinogen binding studies suggested that ERp5 was having an effect on the extended/activated integrin. The native integrin exists predominantly in a bent conformation with closed headpiece. Soaking of the αIIbβ3 integrin headpiece with RGD peptide ligand results in variably extended configurations (*Zhu et al., 2013*). Six intermediate conformations and fully extended conformation with open headpiece have been described. Incubation of RGD-bound αIIbβ3 integrin with ERp5 resulted in significant cleavage of only one of the 24 β3 disulfides: the βI-domain Cys177-Cys184 bond. There was dose-dependent cleavage of the disulfide by ERp5 (*Figure 3F*) and the bond was not cleaved by PDI (*Figure 3G*), indicating selectivity for ERp5. Control RGE peptide that does not bind the integrin did not facilitate cleavage of the disulfide by ERp5 (*Figure 3—figure supplement 2*). As anticipated, redox-inactive ERp5 did not cleave the bond (*Figure 3F*). These results indicate that ERp5 specifically cleaves the βI-domain Cys177-Cys184 disulfide in one or more of the extended/open αIIbβ3 conformations. Approximately 30% of the Cys177-Cys184 disulfide bond in the purified integrin preparation is cleaved by 10-fold molar excess of ERp5 under static conditions, which is a ~20% increase over baseline. The force spectroscopy findings suggest that extent of cleavage is likely to be higher when the integrin is subject to mechanical shearing. ERp5-mediated dissociation of fibrinogen from αIIbβ3 is greatly enhanced when force goes beyond 15 pN and is complete at 40 pN (*Figure 2D*).

The βI-domain Cys177-Cys184 disulfide bond is also cleaved on the platelet surface by platelet ERp5. Washed human platelets were incubated function-blocking anti-ERp5 antibodies or isotype control antibodies and the redox state of the βI-domain disulfide determined in the integrin immunoprecipitated from lysate. Lysis of platelets releases stored ERp5 and fibrinogen that was predicted to mediate cleavage of the Cys177-Cys184 disulfide bond. The Cys177-Cys184 disulfide bond was reduced in ~10% of the integrin population in untreated or control antibody treated platelet lysate (*Figure 3H*), which is in accordance the redox state of this bond in purified platelet αIIbβ3 (*Figure 3D–G*). Incubation of platelets with function-blocking anti-ERp5 antibodies during lysis inhibited reduction of the Cys177-Cys184 bond (p<0.001) (*Figure 3H*).

## The Cys177-Cys184 disulfide has an allosteric configuration

The Cys177-Cys184 disulfide is a –/+RHhook in all crystal structures of the protein, which includes bent, extended, ligand-free and ligand-bound structures (*Supplementary file 2*). Disulfide bonds are classified based on the geometry of the five dihedral angles that define the cystine residue (*Schmidt et al., 2006*). Twenty possible disulfide bond configurations are possible using this classification scheme and all 20 are represented in protein structures. Some disulfide bonds are cleaved in the mature protein to control function (*Hogg, 2003*), the so-called allosteric bonds (*Schmidt et al., 2006*), and these disulfides are increasingly recognized to have one of three configurations (*Butera et al., 2014a*). The –/+RHhook is one of these configurations, along with –RHstaple and –LHhook bonds (*Butera et al., 2014b*). The conformational constraints imposed on the –/+RHhook

and –RHstaple disulfides by topological features stress the bonds via direct stretching of the sulfur-sulfur bond and neighbouring angles (*Schmidt et al., 2006*; *Zhou et al., 2014*). Stretching of sulfur-sulfur bonds increases their susceptibility to cleavage (*Baldus and Gräter, 2012*; *Li and Gräter, 2010*; *Wiita et al., 2006*; *Wiita et al., 2007*), so this internal stress fine tunes bond cleavage and thus the function of the protein in which the bond resides.

## Both catalytic domains of ERp5 are required for substrate specificity

The position of the Cys177-Cys184 disulfide relative to the ligand binding pocket and access to the bond by ERp5 was examined in the crystal structure of the extended holo headpiece (PDB code 2vdo). Cys184 of the bond is surface exposed (*Supplementary file 2*) on a face of the βI-domain that is remote from the ligand binding pocket (*Figure 4A*). This suggests that the Cys184 sulfur atom of the disulfide is attacked by ERp5 to cleave the bond and explains why ERp5 access is not blocked by RGD ligand binding.

The ERp5 N-terminal part consists of two thioredoxin-like domains containing a catalytic dithiol/disulfide in CysGlyHisCys motifs, *a* and *a'*, separated by an *x* segment (*Figure 4B*). These domains are followed by a possible substrate binding domain, *b*. The redox potentials of the *a* (Cys55-Cys58) and *a'* (Cys190-Cys193) catalytic disulfides of ERp5 were determined using differential cysteine alkylation and mass spectrometry. The equilibrium data is shown in *Figure 4B*. The standard redox potentials of the *a* and *a'* domain disulfides of ERp5 are −206 mV and −211 mV, respectively. These redox potentials are about mid-way between the potentials of the PDI (*Bekendam et al., 2016*) and thioredoxin (*Lundström and Holmgren, 1993*) catalytic disulfides.

N- and C-terminal fragments of ERp5 containing a single active-site were tested for cleavage of the Cys177-Cys184 disulfide. Both fragments cleaved the bond with the same efficiency as full-length protein (*Figure 4C*). This is in agreement with the equivalent redox potentials of the active-site dithiols/disulfides (*Figure 4B*). It also indicates that the substrate binding domain of ERp5 is not required for access to and cleavage of the Cys177-Cys184 disulfide. It was possible, though, that separating the two catalytic domains of ERp5 would influence substrate specificity, that is, the disulfide bond or bonds cleaved by ERp5. This was tested by examining the effect of the ERp5 fragments on adhesion of washed human platelets to fibrinogen in the first 4 min of flow at a shear rate of 1000 s$^{-1}$. These conditions were chosen as full-length ERp5 has no effect over this time frame at this shear rate (*Figure 1C*).

As for full-length ERp5 and redox-inactive ERp5, where the active site cysteines of both thioredoxin-like domains are replaced with serines, the N-terminal catalytic domain of ERp5 had no effect on platelet adhesion to fibrinogen under these conditions. In marked contrast, the C-terminal domain enhanced platelet adhesion to fibrinogen by ~50 fold (*Figure 4D*). Large platelet aggregates adhered to the fibrinogen-coated slides (*Figure 4D* inset). The C-terminal domain also enhanced platelet aggregation is response to a PAR-1 agonist, while full-length ERp5 had no effect (*Figure 4E*). These findings indicate that both catalytic domains in full-length ERp5 are required for specificity of cleavage of the Cys177-Cys184 disulfide. The result implies that separating the domains leads to cleavage of other disulfide bonds in the system and different functional effects.

## Cleavage of the βI disulfide results in reduced affinity for fibrinogen due to increased βI-domain flexibility and high stresses at the MIDAS site

An intact Cys177-Cys184 disulfide bond was found to be required for normal fibrinogen binding in a 2004 structure/function study of the disulfide bonds in β3 integrin (*Kamata et al., 2004*). We confirmed this result by measuring binding of soluble or immobilized fibrinogen to BHK cells expressing wild-type or disulfide mutant (C177,184S) αIIbβ3 integrin in the absence or presence of the integrin activator, Mn$^{2+}$. Expression of wild-type (64.8% of cells) and disulfide mutant integrin (82.9% of cells) was confirmed by staining with anti-β3 antibody (*Figure 5A*). Soluble fibrinogen bound to less than 20% of wild-type or disulfide mutant β3 positive cells in the absence of Mn$^{2+}$ (*Figure 5B*). In the presence of Mn$^{2+}$, binding of soluble fibrinogen to αIIbβ3 with a broken Cys177-Cys184 disulfide bound was impaired (p<0.05) compared to wild-type integrin (*Figure 5B*). Cells expressing wild-type integrin adhered to immobilized fibrinogen in the absence and presence of Mn$^{2+}$, whereas

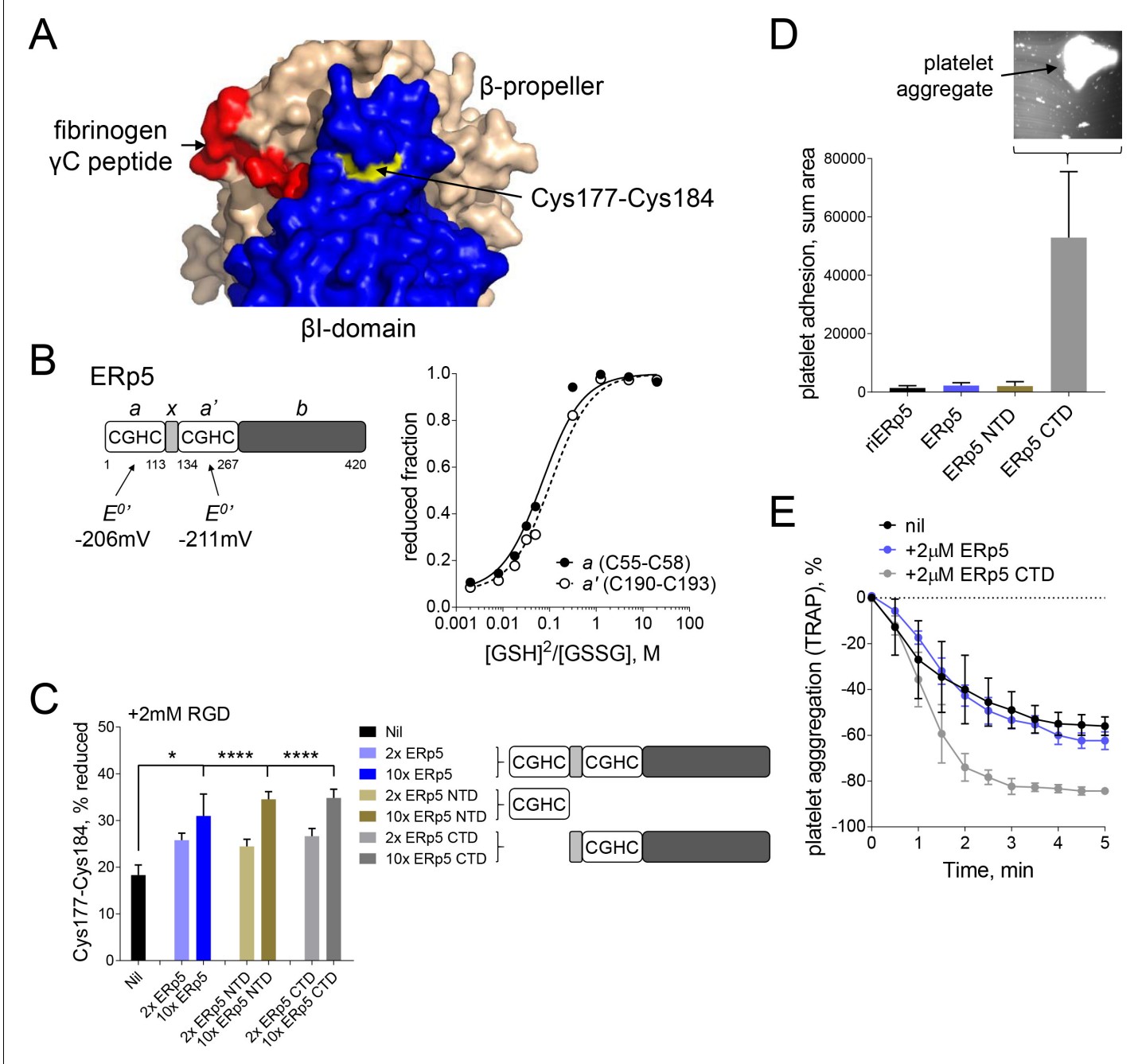

**Figure 4.** Both catalytic domains of ERp5 are required for substrate specificity. (A) Surface structure of the extended holo conformation headpiece (PDB code 2vdo) showing the βI-domain (blue), β-propeller domain of the α2b subunit (wheat) and fibrinogen γC peptide (red). Cys184 of the Cys177-Cys184 disulfide (yellow) is surface exposed in a pocket that is removed from the ligand binding pocket. (B) Redox potentials of the *a* (Cys55-Cys58) and *a'* (Cys190-Cys193) active-site dithiols/disulfides of ERp5. Plots of the fraction of reduced ERp5 as a function of the ratio of GSH to GSSG. The lines represent the best non-linear least squares fit of the data to *Equation 1*. The calculated equilibrium constants were used to determine the standard redox potentials from *Equation 2*. Data points and errors are the mean of 2–4 peptides encompassing the active site cysteine residues. (C) Cleavage of the Cys177-Cys184 disulfide bond by 2- or 10-fold molar excess of full-length ERp5, ERp5 N-terminal domain, or ERp5 C-terminal domain. The bars and errors (1 SD) are for 2–6 measurements. *p<0.05, ****p<0.001; assessed by unpaired, two-tailed Student's t-test. (D) Platelet adhesion to fibrinogen at 4 min in the absence or presence of 2 µM redox inactive ERp5 (riERp5), ERp5 or the N- or C-terminal ERp5 domains at a fluid shear rate of 1000 s⁻¹. The inset is a representative image of platelet adhesion to immobilized fibrinogen in the presence of 2 µM ERp5 CTD. The bars and errors (1 SD) are from three measurements each from six different healthy donor platelets. (E) Aggregation of washed platelets activated with the PAR-1 agonist, TRAP (7 µM), in the absence or presence of 2 µM ERp5 or ERp5 CTD. The data points and errors (1 SEM) are from three different healthy donor platelets.
DOI: https://doi.org/10.7554/eLife.34843.009

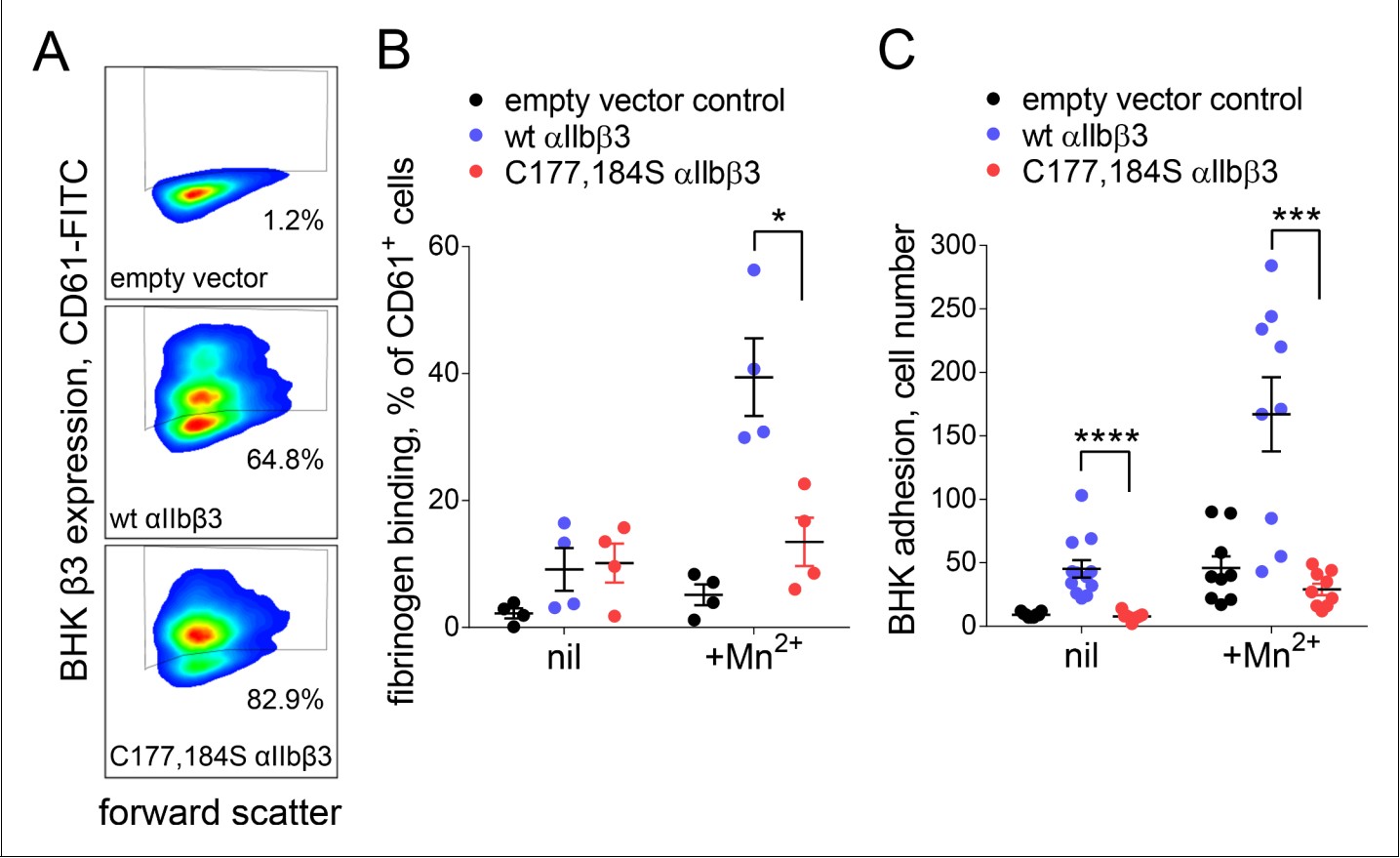

**Figure 5.** Cleavage of the βI domain disulfide impairs fibrinogen binding. (A) Expression of wild-type (64.8% of cells) and β3 C177,184S disulfide mutant αIIbβ3 integrin (82.9% of cells) on BHK cells from staining with fluorescene-conjugated anti-CD61 antibody and flow cytometry. Transfection with empty vector served as negative control. (B) Binding of soluble fibrinogen-AF647 (0.1 mg/mL) to BHK cells expressing wild-type or C177,184S mutant αIIbβ3 integrin. Binding is expressed as the % of CD61+ cells that bound fibrinogen. Data points and errors (SD) are from four separate experiments. (C) Adhesion of BHK cells expressing wild-type or C177,184S mutant αIIbβ3 integrin to immobilized fibrinogen. Experiments were performed in triplicate, three fields were analyzed per well and errors are SD. *p<0.05, ***p<0.005, ****p<0.001; assessed by unpaired, two-tailed Student's t-test.

DOI: https://doi.org/10.7554/eLife.34843.010

adherence of the disulfide mutant integrin was severely impaired in both conditions (p<0.0001 in the absence of $Mn^{2+}$ and p<0.001 in the presence of $Mn^{2+}$) (*Figure 5C*).

The structural changes in the oxidized and reduced states of the αIIbβ3 integrin headpiece that underpin the impaired affinity for fibrinogen were examined by Molecular Dynamics simulations. Three different αIIbβ3 starting structures were analyzed; the bent conformation (PDB code 3fcs), extended apo conformation (PDB code 3fcu), and extended holo conformation that includes the fibrinogen γC peptide (PDB code 2vdo). We find disulfide bond reduction to render the whole head-piece more flexible in all three cases, as reflected by a wider distribution of conformations along conformational modes as obtained from Principal Component Analysis (*Figures 6A* and *Figure 6— figure supplement 1*). The strongest effect is observed for the extended holo structure compared to bent and extended apo structures (compare *Figure 6A* with *Figure 6—figure supplement 1*). The allosteric network originating from disulfide bond reduction, measured by force distribution analysis, emanates from the metal binding sites into the domain periphery, as shown by using decreasing force reduction, whereas the force differences in the β-propeller domain are minor. At lower forces, however, this domain is also allosterically reached. In more detail, the network involves both cysteines, the critical residue for ligand binding D119, as well as residues involved in ion positioning, such as D217 and N214 (*Figure 6E* and *Figure 6—figure supplement 1*).

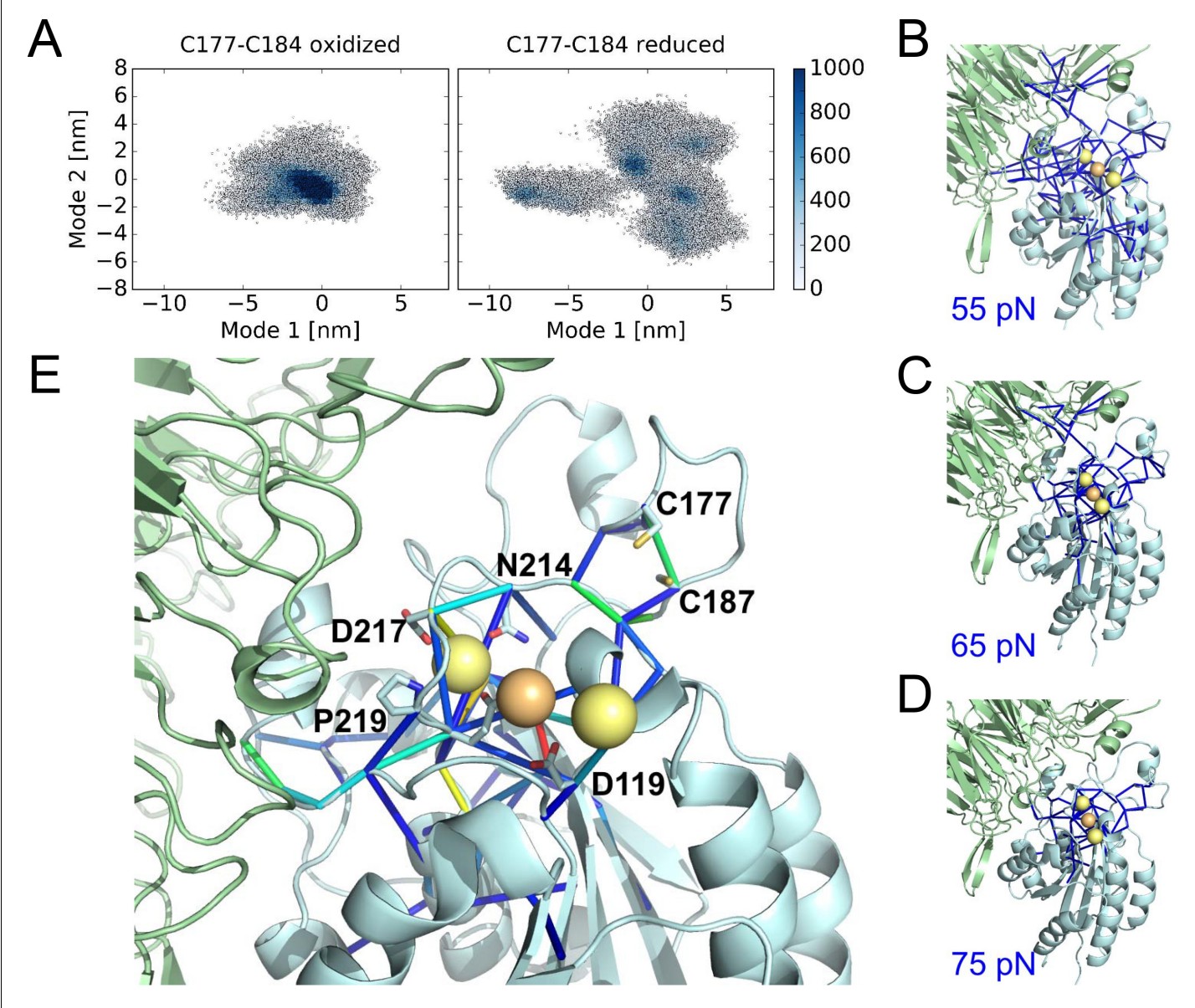

**Figure 6.** Cleavage of the βI disulfide increases βI-domain flexibility and stresses at the MIDAS site. (**A**) Conformational distribution of the oxidized (left) and reduced (right) αIIbβ3 integrin apo headpiece along two major conformational modes as obtained from MD simulations, starting from the extended apo conformation (PDB code 3fcu). The color code shows the density of conformations, ranging from low (white) to high (dark blue) density (in arbitrary units). Disulfide reduction increases the covered area and, thus, the conformational fluctuations. (**B–D**) Allosteric signaling network upon reduction of the Cys177-Cys184 disulfide bond. Differences in Fpair-wise force between the oxidized and reduced integrin headpiece are shown as blue sticks for the indicated force cut-off values. Calcium and magnesium ions are presented as yellow and orange spheres, respectively, the β-propeller domain as the green cartoon and the βI-domain as the cyan cartoon. (**E**) Details of the Fpair-wise force difference network around the metal-binding sites in the βI domain. Stress intensity is indicated by the color spectrum as used for b-factors (spectrum ranging from 75 pN – 399 pN).
DOI: https://doi.org/10.7554/eLife.34843.011

The following figure supplement is available for figure 6:

**Figure supplement 1.** Cleavage of the βI disulfide results in increased βI domain flexibility and high stresses at the MIDAS site in the bent apo and extended holo αIIbβ3 structures.
DOI: https://doi.org/10.7554/eLife.34843.012

## Discussion

ERp5 secreted by activated platelets binds to the β3 subunit of platelet αIIbβ3 integrin (*Jordan et al., 2005*; *Passam et al., 2015*). We now report that ERp5 cleaves the βI-domain Cys177-Cys184 disulfide bond nearby the fibrinogen binding pocket of extended activated integrin that results in release of fibrinogen (*Figure 7*). Two coupled events control cleavage of the disulfide bond: ligand binding and mechanical force. RGD-ligand binding to the integrin and shear force facilitate ERp5 reduction of the disulfide. Stretching of sulfur-sulfur bonds, either by internal pre-stress or external forces, increases their susceptibility to cleavage (*Baldus and Gräter, 2012*; *Li and Gräter, 2010*; *Wiita et al., 2006*; *Wiita et al., 2007*). Our data suggest that ERp5 cleavage of the disulfide is enabled by ligand- and force-dependent stretching of the sulfur-sulfur bond. However, the experimental data are also consistent with a force-dependent, ligand-bound conformation that provides enhanced access of the disulfide to ERp5. This force-coupled ligand binding redox event is an intriguing example of mechano-chemically coupled catalysis (*Neumann and Tittmann, 2014*). Our findings are also of significance in understanding how platelets harness force to balance haemostasis and thrombosis functions.

The Cys177-Cys184 disulfide bond is exposed to solvent and accessible to ERp5 on a face of the βI-domain that is not involved in ligand binding. Cleavage of the disulfide bond, however, results in long-range allosteric effects within the βI-domain of αIIbβ3, mainly affecting the metal-binding sites, along with a higher conformational mobility of the whole βI-domain. Interestingly, we do not observe

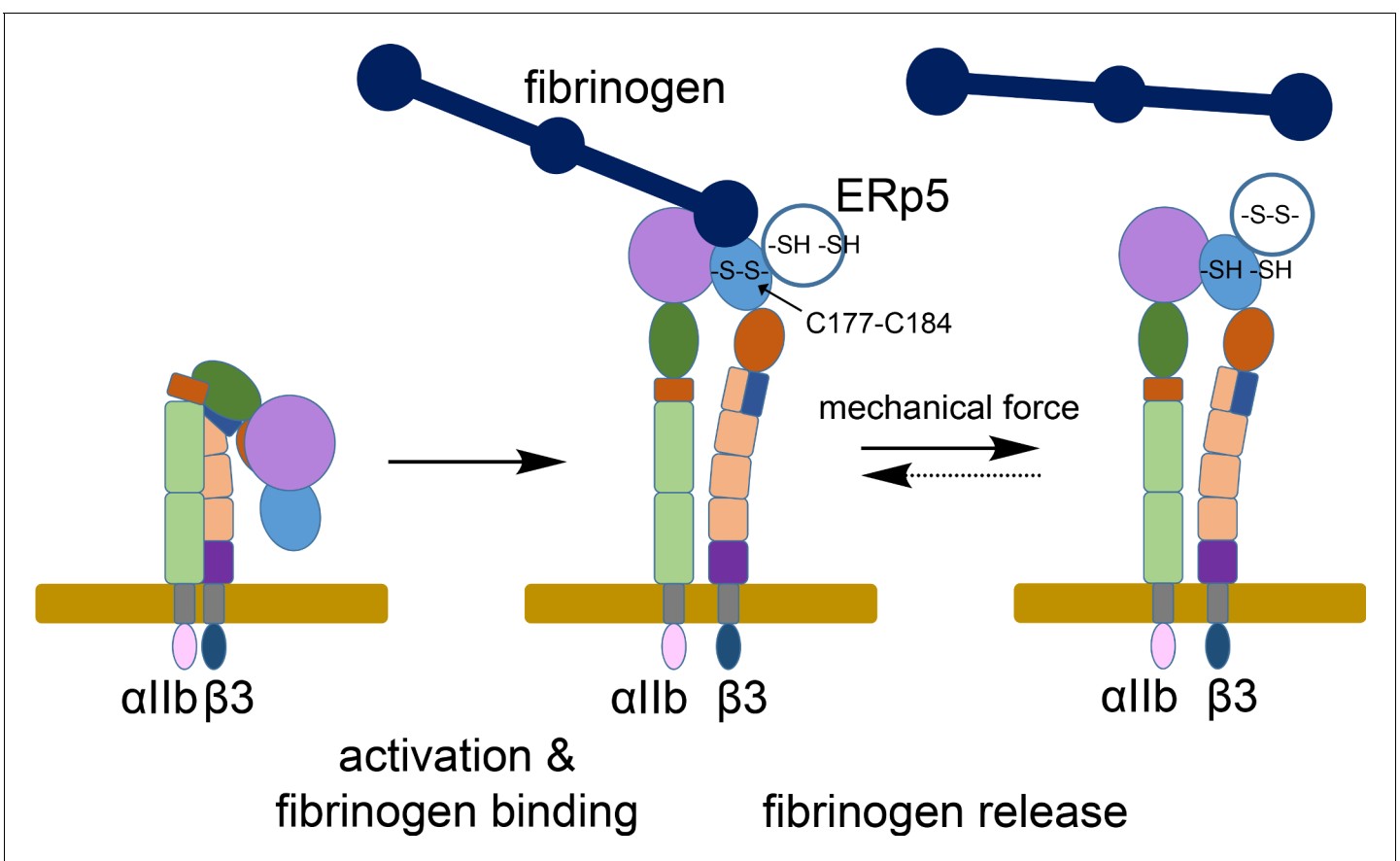

**Figure 7.** ERp5 regulation of fibrinogen release from activated αIIbβ3 integrin. Activation of αIIbβ3 integrin results in transition from a bent (low affinity) to an extended (high affinity) structure. While much is known about the structure and function of resting and activated integrins, little is known about how integrins disengage from their ligands. Our findings indicate that ERp5 regulates fibrinogen release from extended αIIbβ3 integrin by reducing the βI domain Cys177-Cys184 disulfide bond (light blue oval). Cleavage of the disulfide is controlled by both ligand binding and mechanical force. It is feasible that the disulfide bond could reform and the integrin reengage fibrinogen (dotted arrow).
DOI: https://doi.org/10.7554/eLife.34843.013

significant changes of α-helix 7, which has been previously shown to regulate the affinity of the integrin for (RGD) ligands (*Luo and Springer, 2006*). Our data instead suggest that the increased mobility due to disulfide bond scission leads to local effects around the RGD-binding MIDAS site, which in turn are directly responsible for the observed affinity reduction of αIIbβ3 for fibrinogen upon Cys177-Cys184 reduction by ERp5. We note that we cannot exclude allosteric effects on the α-helix seven on longer time scales.

Both the N- and C-terminal catalytic domains of ERp5 are required for specificity of cleavage of the Cys177-Cys184 disulfide. Although the individual domains, when expressed and tested separately, were found to cleave the Cys177-Cys184 disulfide with comparable efficiency, removal of the N-terminal thioredoxin-like domain from ERp5 resulted in a fragment that markedly promoted platelet clumping on a fibrinogen-coated surface under fluid shear and enhanced PAR-1 mediated platelet aggregation. Interestingly, the C-terminal thioredoxin domains of ERp57 and PDI also potentiate platelet aggregation (*Zhou et al., 2015*; *Wang and Essex, 2017*). The PDI C-terminal thioredoxin domain mediates P-selectin expression and ATP secretion in a αIIbβ3-independent manner. The effects of the isolated thioredoxin domains are likely the result of cleavage of other disulfide bond(s) in platelet proteins. These findings highlight the possibility that proteolytic processing of the oxidoreductases in the circulation may generate fragments with independent activities. They also suggest that care that needs to be taken when targeting oxidoreductases for development of new anti-thrombotics (*Flaumenhaft et al., 2015*). Inhibiting different aspects of the factors may have unintended consequences.

Our studies of ERp5's role in thrombus formation in vivo (*Passam et al., 2015*) and ERp5 effects on platelet adhesion under flow shown herein, indicate that ERp5 is involved in propagation of the thrombus and not thrombus initiation. ERp5 accumulates in the developing thrombus (*Passam et al., 2015*), which may be a mechanism to limit thrombus growth by shifting the balance between fibrinogen cross-linking of platelet αIIbβ3 and fibrinogen dissociation from this receptor. Testing this theory in vivo will be complicated by the likely possibility that ERp5 has more than one substrate in the thrombus. For instance, our preliminary results indicate that ERp5 significantly enhances binding of platelets to von Willebrand factor.

The βI Cys177-Cys184 disulfide bond is conserved in 7 of the 8 β integrins, or 23 of the 24 vertebrate integrins (*Supplementary file 3*). β4 of the α6β4 laminin receptor is the only β integrin that does not contain the bond. Notably, PDI and ERp57, like ERp5, have been shown to bind to β3 integrins. PDI binds to β3 integrins in the thrombus (*Cho et al., 2012*; *Kim et al., 2013*) and αvβ3 integrin on endothelial cells (*Swiatkowska et al., 2008*), while ERp57 binds to platelet surface β3 integrin (*Holbrook et al., 2012*; *Wang et al., 2013*). It is possible that different oxidoreductases regulate de-adhesion of different integrins by cleaving the βI-domain disulfide bond. Neutrophil PDI, for instance, modulates ligand binding to αMβ2 integrin and neutrophil recruitment during venous inflammation (*Hahm et al., 2013*). Future studies will show to what extent mechano-redox regulation is a mechanism at play beyond β3/ERp5 to control other integrins and other protein-protein interactions.

## Materials and methods

### αIIbβ3 integrin activation on human platelets

All procedures involving collection of human blood from healthy volunteers were in accordance with St George Hospital Human Ethics Committee (HREC 16/009), Human Research Ethics Committee of the University of Sydney (HREC 2014/244) and the Helsinki Declaration of 1983. Whole blood was drawn into ACD-A tubes (BD Vacutainer) and platelet rich plasma collected by centrifugation at 200 g for 20 min at room temperature. Following addition of 1 μM prostaglandin E1, platelets were collected by centrifugation at 800 g for 20 min, washed with Hepes Tyrodes glucose buffer (20 mM Hepes, 134 mM NaCl, 0.34 mM $Na_2HPO_4$, 2.9 mM KCl, 12 mM $NaHCO_3$, 1 mM $MgCl_2$, 5 mM glucose, pH 7.4) and resuspended in the same buffer at a concentration of 300,000–400,000 per μL. Recombinant ERp5 and protein disulfide isomerase (PDI) were produced in *E. coli* as described (*Passam et al., 2015*). Plasmids for N- and C-terminal domains of ERp5 were from Thomas Spies, Fred Hutchinson Cancer Research Centre, USA. Platelets ($10^6$ in 0.1 mL) were incubated with ERp5 (2 μM) in the absence or presence of ADP (20 μM) for 2 min at room temperature. Platelets were

stained with fluorescene-conjugated PAC-1 antibody (RRID:AB_2230769) (0.5 µg/ml) for 30 min at 25°C, washed and fixed with 1% paraformaldehyde, and binding measured by flow cytometry on a BD FACS Canto II.

## Platelet adhesion assays in flow chambers

The assembly and function of the platelet flow chambers are described in more detail in Bio-protocol (*Dupuy et al., 2019*). Microchannels of Vena8 Fluor Biochip (Cellix Ltd) were coated with 10 µL of 20 µg/mL human fibrinogen overnight at 4°C in a humidified box, blocked with 10 µL of 0.1% bovine serum albumin in phosphate buffered saline (PBS) for 1 hr at room temperature and washed with 40 µL of PBS. Washed platelets were prepared as above, labeled with 1 µg/mL calcein (Thermo Fisher) and injected by Mirus NanoPump into the channels at shear rates of 1000 s$^{-1}$ and 3000 s$^{-1}$ (flow rates of 40 and 120 µl/min, respectively) within 3 hr from blood collection. Adhesion of platelets was monitored in real time with images captured via an ExiBlu CCD camera (Q imaging, Canada) connected to an AxioObserver A1 Inverted Epi-Fluorescence microscope (Zeiss, Germany). Images were captured using the accompanying VenaFlux 2.3 imaging software. Images were analyzed at positions 2, 4 and 6 (located at 6, 14 and 22 mm from the entry site of blood) of the microchannels at 1 min intervals (since initiation of flow) using the ImagePro Premier 64-bit software. These positions are representative of flow nearest, mid-way and furthest from the entry of blood into the channel. Data were exported into Excel and area coverage by platelets was calculated for each position. Results are from 4 to 9 donors for each experiment.

## Platelet aggregation assays

Aggregation studies were performed with washed platelets prepared as described (*Zhou et al., 2015*; *Wang et al., 2013*). Washed platelets were resuspended in Hepes Tyrodes glucose buffer at 300,000 per µL. Platelets were incubated with ERp5 proteins for 3 min and aggregation initiated with 7 µM TRAP-6 (Roche). Aggregation was measured by light transmission using a ChronoLog Aggregometer (Model 560-Ca). Results are from three healthy donor platelets and expressed as % aggregation over time.

## Dynamic force spectroscopy assays

Biomembrane Force Probe (BFP) experiments were performed as we previously described (*Ju et al., 2013*; *Ju et al., 2015a*; *Ju et al., 2015b*). Briefly, the BFP utilizes two micropipettes, one aspirating a biotinylated human red blood cell (RBC) with a glass bead to serve as a force transducer, termed 'probe'. The bead is attached on the RBC apex via biotin-SA interaction. The probes spring constant, set to 0.25–0.3 pN/nm, is determined by the aspiration pressure and the radii of the micropipette and the RBC-bead contact area. The other micropipette aspirates a second bead, termed 'target'. The probe and target beads are respectively coupled with purified human fibrinogen and αIIbβ3 with maleimide-PEG3500-NHS (JenKem, USA) in carbonate/bicarbonate buffer (pH 8.5). The force spectroscopy traces are obtained by measuring the RBC-bead deflection from the probe beads edge tracking. Bond formation/dissociation and force application are enabled and monitored in controlled BFP touch cycles (~2.5 s each). Other details are found in published protocols (*Chen et al., 2015*; *Ju et al., 2017*). RBCs are isolated, biotinylated with Biotin-PEG3500-NHS (Jen-Kem, TX) and stored (up to 2 weeks) for BFP experiments.

In each cycle, the αIIbβ3-bearing target bead is driven to approach and contact the fibrinogen-probe bead with a 20-pN compressive force for a certain contact time (0.2 s) that allows for bond formation. The target is then retracted at a constant speed (3.3 µm/s) for bond detection. During the retraction phase, a 'bond' event is signified by a tensile force. Conversely, no tensile force indicates a 'no-bond' event. For the adhesion frequency assay, 'bond' and 'no-bond' events are enumerated to calculate an adhesion frequency in 50 repeated cycles for each probe–target pair. For a force-clamp assay to measure bond lifetimes, upon detection of 'bond' event in a similar BFP cycle, a feedback loop pauses the retraction at a desired clamped force (5–60 pN) until bond dissociation. After that, the target is recoiled to the original position to complete the cycle. Lifetimes are measured from the instant when the force reaches the desired level to the instant of bond dissociation.

## Redox states of the β3 integrin disulfide bonds

The redox states of 24 of the 28 β3 integrin disulfide bonds was measured in isolated human platelet β3 integrin (Abcam). Unpaired cysteine thiols in β3 integrin were alkylated with 5 mM 2-iodo-N-phenylacetamide ($^{12}$C-IPA, Cambridge Isotopes) for 1 hr at room temperature, the protein resolved on SDS-PAGE and stained with colloidal coomassie (Sigma). The β3 band was excised, destained, dried, incubated with 40 mM dithiothreitol and washed. The fully reduced protein was alkylated with 5 mM 2-iodo-N-phenylacetamide where all six carbon atoms of the phenyl ring have a mass of 13 ($^{13}$C-IPA, Cambridge Isotopes). The gel slice was washed, dried and deglycosylated using 5 units PNGase F (Sigma), before digestion of β3 integrin with 12.5 ng/μl of chymotrypsin (Roche) in 25 mM $NH_4CO_2$ and 10 mM $CaCl_2$ for 4 hr at 37°C followed by digestion with 12.5 ng/μl of trypsin overnight at 25°C. Peptides were eluted from the slices with 5% formic acid, 50% acetonitrile. Liquid chromatography, mass spectrometry and data analysis were performed as described (*Chiu et al., 2014*; *Cook et al., 2013*). Sixty-eight peptides encompassing disulfide Cys residues were resolved and quantified (*Figure 3—figure supplement 1*). The levels of the different redox forms of the cysteines was calculated from the relative ion abundance of peptides labelled with $^{12}$C-IPA and/or $^{13}$C-IPA. To calculate ion abundance of peptides, extracted ion chromatograms were generated using the XCalibur Qual Browser software (v2.1.0; Thermo Scientific). The area was calculated using the automated peak detection function built into the software. More detailed protocol for differential cysteine labelling and mass spectrometry quantification is described in Bio-protocol (*Chiu, 2019*).

## Redox potential measurements

The redox potentials of the *a* (Cys53-Cys56) and *a'* (Cys397-Cys400) active-site dithiols/disulfides of ERp5 were determined by differential cysteine alkylation and mass spectrometry. Recombinant ERp5 (5 μM) was incubated in argon-flushed phosphate-buffered saline containing 0.1 mM EDTA, 0.2 mM oxidized glutathione (GSSG, Sigma) and various concentrations of reduced glutathione (GSH, Sigma) for 18 hr at room temperature to allow equilibrium to be reached. Microcentrifuge tubes were flushed with argon prior to sealing to prevent oxidation by ambient air during the incubation period. Unpaired cysteine thiols in ERp5 were alkylated with 5 mM $^{12}$C-IPA for 1 hr at room temperature. The proteins were resolved on SDS-PAGE and stained with SYPRO Ruby. The ERp5 bands were excised, destained, dried, incubated with 100 mM dithiothreitol (DTT) and washed. The fully reduced proteins were alkylated with 5 mM $^{13}$C-IPA and the gel slices washed and dried before digestion of proteins with 12 ng/μL of chymotrypsin (Roche) in 25 mM $NH_4CO_2$ overnight at 25°C. Peptides were eluted from the slices with 5% formic acid, 50% acetonitrile. Liquid chromatography, mass spectrometry and data analysis were performed as described (*Cook et al., 2013*).

The fraction of reduced active-site disulfide bond was measured from the relative ion abundance of peptides containing $^{12}$C-IPA and $^{13}$C-IPA. To calculate ion abundance of peptides, extracted ion chromatograms were generated using the XCalibur Qual Browser software (v2.1.0; Thermo Scientific). The area was calculated using the automated peak detection function built into the software. The ratio of $^{12}$C-IPA and $^{13}$C-IPA alkylation represents the fraction of the cysteine in the population that is in the reduced state. The results were expressed as the ratio of reduced to oxidized protein and fitted to *Equation 1*:

$$R = \frac{\left(\frac{[GSH]^2}{[GSSG]}\right)}{K_{eq} + \left(\frac{[GSH]^2}{[GSSG]}\right)} \tag{1}$$

where $R$ is the fraction of reduced protein at equilibrium and $K_{eq}$ is the equilibrium constant. The standard redox potential ($E^{0'}$) of the ERp5 active-site disulfides were calculated using the Nernst equation (*Equation 2*):

$$E^{0'} = E^{0'}_{GSSG} - \frac{RT}{2F} \ln K_{eq} \tag{2}$$

using a value of −240 mV for the standard redox potential of the GSSG disulfide bond.

## Immunoprecipitation of platelet β3

Washed platelets ($10^6$ in 0.1 mL) were prepared as above, labeled with $^{12}$C-IPA (5 mM) for 1 hr at room temperature, centrifuged at 2000 g for 10 min and washed with Hepes Tyrodes glucose buffer. Platelets were lysed with 0.1 mL of 2% NP40, 30 mM Hepes, 150 mM NaCl, 2 mM EDTA, pH 7.4 buffer containing proteinase inhibitor cocktail, and lysate was collected after centrifugation at 10,000 g for 20 min. Lysate (2 mg) and AP3 antibody (RRID:AB_2056630) (40 μg) were mixed in 0.5 ml of IP/lysis buffer (Pierce) and rotated overnight at 4°C. αIIbβ3 integrin was collected on 80 μl of protein A/G agarose with rotation for 2 hr at 25°C. The beads were washed three times with IP lysis/ buffer and three times with PBS. $^{12}$C-IPA (5 mM) was added to the beads and incubated with rotation for 1 hr at 25°C. The integrin was eluted from the beads with 0.1 M glycine, the pH neutralized with 0.1 M Tris, pH 9.5 buffer and the β3 subunit resolved on SDS-PAGE and processed as above.

cDNA of β3 was cloned in pcDNA3 vector (carrying the neomycin resistance gene) and cDNA of αIIb was cloned into pCEP4 vector (carrying the hygromycin resistance gene) as previously described (*Mor-Cohen et al., 2008*). The β3 C177,184S mutant was created in the pcDNA β3 vector by site-directed mutagenesis using the QuikChange kit from Stratagene. Verification of mutations was confirmed by DNA sequencing. Plasmids were linearized using PVUI for β3/pcDNA3 and AvrII for αIIb/ pCEP4. BHK cells, grown in DMEM supplemented with 2 mg/ml L-glutamine and 5% FCS, were co-transfected with 1 μg of wild type or mutant pcDNA/β3 and 1 μg of pCEP4/αIIb with lipofectamine. Cells were also transfected with empty pcDNA and pCEP4 vectors as negative control. Transfected cells were grown in media containing 0.5 mg/ml hygromycin and 0.5 mg/ml G418. Two separate transfections for wild type and mutant construct were performed. Cells were sorted for comparable expression of the integrin by staining with fluorescene-conjugated anti-CD61 antibody (RRID:AB_ 929170) and flow cytometry.

For fibrinogen binding assays, near-confluent transfected cells were washed twice with phosphate-buffered saline (PBS) and detached by adding 1 mM EDTA. Cells were suspended in DMEM, pelleted and washed twice before resuspending $0.5 \times 10^6$ in 0.1 mL PBS supplemented with 1 mM $MgCl_2$, 1 mM $CaCl_2$ without or with 1 mM $MnCl_2$. Cells were incubated wth 10 μg/mL CD61-APC and 0.1 mg/mL fibrinogen-FITC for 30 min at 25°C, washed with PBS, fixed with 1% paraformaldehyde, washed twice more with PBS and binding measured by flow cytometry on a BD FACS Canto II. Binding is expressed as the % of CD61+ cells that bound fibrinogen. For cell adhesion assays, near confluent cells were washed in PBS and detached by adding 5 mM EDTA. Cells were suspended to $10^6$ cells/mL in 0.5 mL of 20 mM HEPES pH 7.4 buffer containing 0.1 M NaCl$_2$, 1 mM CaCl$_2$,1 mM MgCl$_2$ without or with 1 mM MnCl$_2$. Flat bottom 96 well MaxiSorp Nunc-Immuno plates (Thermo Fisher) were coated overnight with 40 μg/mL fibrinogen and blocked for two hours with 10 mg/mL denatured BSA. Cells, 100 μL of $10^6$ cells/mL, were added to the wells, allowed to adhere for 2 hr at 37°C, and then gently washed three times with 200 μL PBS. The wells were overlayed with 100 μL PBS and imaged at 10x magnification using an inverted light microscope. Experiments were performed in triplicate and three images were taken per well. The number of attached cells were counted using ImageJ and a custom macro written for this purpose. Cell attachment is represented in absolute numbers.

## Molecular dynamics simulations

The crystal structures of the bent conformation of αIIbβ3 (PDB code 3fcs [*Zhu et al., 2008*]), extended apo conformation of αIIbβ3 (PDB code 3fcu [*Zhu et al., 2008*]), and extended holo conformation of αIIbβ3 (PDB code 2vdo [*Springer et al., 2008*]) were prepared by separating the headpiece from the other domains (β-propeller residues 1–452, βI-domain residues 105–353, and 14 residue bound-peptide in the holo). The termini were capped in PyMOL version 1.8 (*Schrodinger LLC, 2015*) (acetyl group on N-terminus, N-methylaminly group on C-terminus), sugars were neglected and calcium and magnesium ions were kept in the system. The protonation states of the amino acids were calculated using PROPKA (*Rostkowski et al., 2011*). A dodecahedric box, leaving at least 1.5 nm distance between protein and box boundaries, was considered. The box was filled with water and the total system charge was neutralized with sodium and chloride ions (150 mM). For calcium and magnesium ions the default Amber parameters were used, for sodium and chloride the optimized Joung parameters (*Joung and Cheatham, 2008*) were considered, and TIP3P (*Jorgensen et al., 1983*) was used as water model. MD simulations of each redox state (C177-C184

in the βI-domain oxidized and reduced) were performed with GROMACS 2016 (*Abraham et al., 2015*), using the Amber99sb*-ILDN force field for the protein (*Lindorff-Larsen et al., 2010*). Energy minimization was performed using the steepest-descent algorithm, followed by 500 ps of NVT-ensemble equilibration (Berendsen thermostat (*Berendsen et al., 1984*) with τ = 0.1 ps) with position restraints on the protein (restraint force constant = 1000 kJ mol$^{-1}$ nm$^2$) while random starting velocities were assigned to each trajectory. Next, 1 ns of NPT-ensemble equilibration with position restraints on the protein was performed. The LINCS algorithm (*Hess, 2008*) was used to constraint bonds involving protein hydrogen atoms, while SETTLE (*Miyamoto and Kollman, 1992*) was used to constraint both bonds and angles of water molecules, enabling an integration time step of 2 fs.

The temperature was kept constant at 300 K by coupling the system to the V-rescale (*Berendsen et al., 1984*; *Bussi et al., 2007*) thermostat with τ = 0.1 ps. The pressure was kept constant at 1 bar coupling isotropically the system to a Parinello-Rahman barostat (*Parrinello and Rahman, 1981*), with τ = 5 ps and compressibility $4.5 \times 10^{-5}$ bar$^{-1}$. Lennard-Jones interactions were calculated using a cut-off of 1 nm and long-range electrostatics were calculated by particle-mesh Ewald summation (*Darden et al., 1993*). The Verlet-buffer scheme was employed to treat the non-bonded neighbour interactions. For each of the three structures and each redox state, five independent trajectories were calculated, in production runs of 200 ns in length each, while saving system coordinates every 10 ps. From each run the first 50 ns were accounted as equilibration time and discarded for subsequent analysis, resulting in 750 ns of cumulative simulation time per redox state for each of the starting structures.

Principal Component Analysis (*Amadei et al., 1993*), consisting in the calculation and diagonalization of the atomic-position covariance matrix, was performed using the GROMACS 2016 analysis tools, on the least-squares fitted trajectories of the backbone atoms. Eigenvectors of the covariance matrix were calculated of all ten concatenated runs (both oxidized and reduced). Each concatenated trajectory of each redox states was then individually projected along the first two eigenvectors, which accounted for approximately 30% of the collective motions.

Force distribution analysis (FDA) of MD simulations provides insight into mechanical signal propagation within a protein of interest upon an external perturbation (*Stacklies et al., 2011*), such as the reduction of the Cys177-Cys184 bond in the βI-domain of αIIbβ3 by ERp5 investigated here. FDA was performed on the trajectories of both redox states (Cys177-Cys184 oxidized and reduced), using the FDA implementation in GROMACS 5.0.7 (*Stacklies et al., 2011*) and the corresponding FDA-tools 1.0. The pairwise inter-residue forces Fij between residues i and j, were calculated from each frame. Pairwise forces within the atoms of the protein were used, whereas forces from water and ions were neglected. The time-average <Fij> was computed for each redox form and pairs of residues for which <Fij(reduced)> - <Fij(oxidized)> was larger than specified threshold cut-off values were presented.

## αIIbβ3 integrin modeling

To model the open structure of the complete αIIbβ3 integrin ectodomain, the bent configuration (PDB code 3fcs [*Zhu et al., 2008*]) was used as the starting structure. In order to open the structure, two peptide bonds were broken: residues 600–601 in chain A to simulate the opening of the hinge between calf1 and the thigh ('α knee'), and residues 475–476 in chain B to simulate the opening of the hinge between EGF-1 and EGF-2 ('β knee'). The opened model was structurally aligned to its appropriate domains in the opened model structure of the αVβ3 integrin ectodomain (PDB code 3ije [*Zucker et al., 2016*; *Xiong et al., 2009*]). Missing residues that were not solved in 3fcs (746–774 and 840–873 in chain A, and 75–78 and 477–482 in chain B) were modelled using Modeller (*Webb and Sali, 2016*) with 3ije as a template.

### Statistics

Parametric unpaired two-tailed *t* test was used to evaluate differences between groups. Statistical results are reported as p values < 0.05, <0.01, <0.005 or <0.001.

## Acknowledgements

This study was supported by grants from the National Health and Medical Research Council of Australia (PJH), St George and Sutherland Medical Research Foundation (FP), Royal College of

Pathologists Foundation Kanematsu/Novo Nordisk Research Award (FP and LJ), Helen and Robert Ellis Postdoctoral Fellowship and Tony Basten Postdoctoral Fellowship from the Sydney Medical School Foundation (JC), Diabetes Australia Research Trust grant G179720 and Sydney Medical School early-career researcher kickstart grant (LJ), National Heart Foundation of Australia Postdoctoral Fellowship (101285) (LJ), Deutsche Forschungsgemeinschaft (research unit FOR 1543 to CA-S, KK, and FG), the University of Los Andes (CA-S), Heidelberg University (CA-S), and the Klaus Tschira Foundation (FG).

## Additional information

### Funding

| Funder | Grant reference number | Author |
| --- | --- | --- |
| Royal College of Pathologists of Australasia | Kanematsu/Novo Nordisk Research Award | Freda Passam |
| St George and Sutherland Medical Research Foundation | | Freda Passam |
| Sydney Medical School Foundation | Helen and Robert Ellis Postdoctoral Fellowship | Joyce Chiu |
| Sydney Medical School Foundation | Tony Basten Postdoctoral Fellowship | Joyce Chiu |
| National Heart Foundation of Australia | Postdoctoral Fellowship 101285 | Lining Ju |
| Diabetes Australia Research Trust | Grant G179720 | Lining Ju |
| Sydney Medical School | Early-career researcher kickstart grant | Lining Ju |
| Deutsche Forschungsgemeinschaft | Research Unit FOR 1543 | Katra Kolšek Camilo Aponte-Santamaría Frauke Gräter |
| University of Los Andes | | Camilo Aponte-Santamaría |
| Universität Heidelberg | | Camilo Aponte-Santamaría |
| Klaus Tschira Stiftung | | Frauke Gräter |
| National Health and Medical Research Council | Research Fellowship 1110219 | Philip J Hogg |

The funders had no role in study design, data collection and interpretation, or the decision to submit the work for publication.

### Author contributions

Freda Passam, Conceptualization, Formal analysis, Supervision, Investigation, Writing—review and editing; Joyce Chiu, Conceptualization, Formal analysis, Investigation, Writing—review and editing; Lining Ju, Formal analysis, Investigation, Writing—review and editing; Aster Pijning, Zeenat Jahan, Formal analysis, Investigation; Ronit Mor-Cohen, Adva Yeheskel, Conceptualization, Investigation; Katra Kolšek, Lena Thärichen, Data curation, Investigation; Camilo Aponte-Santamaría, Data curation, Investigation, Writing—review and editing; Frauke Gräter, Conceptualization, Supervision, Project administration, Writing—review and editing; Philip J Hogg, Conceptualization, Resources, Data curation, Formal analysis, Supervision, Funding acquisition, Writing—original draft, Project administration, Writing—review and editing

### Author ORCIDs

Camilo Aponte-Santamaría (iD) https://orcid.org/0000-0002-8427-6965
Philip J Hogg (iD) http://orcid.org/0000-0001-6486-2863

## Ethics

Human subjects: All procedures involving collection of human blood from healthy volunteers were in accordance with the St George Hospital Human Ethics Committee (HREC 16/009), Human Research Ethics Committee (HREC) (Project number 2014/244) of the University of Sydney, and the Helsinki Declaration of 1983.

## Decision letter and Author response

Decision letter https://doi.org/10.7554/eLife.34843.020
Author response https://doi.org/10.7554/eLife.34843.021

## Additional files

### Supplementary files

• Supplementary file 1. List of β3 integrin cysteine containing peptides detected by mass spectrometry.
DOI: https://doi.org/10.7554/eLife.34843.014

• Supplementary file 2. Structural features of the β3 integrin Cys177-Cys184 disulfide bond.
DOI: https://doi.org/10.7554/eLife.34843.015

• Supplementary file 3. The βI Cys177-Cys184 disulfide bond is conserved in 7 of 8 β integrins. Phylogenetic tree for the integrin β-subunit Cys177-Cys184 disulfide bond. The cysteines forming the disulfide are highlighted in yellow.
DOI: https://doi.org/10.7554/eLife.34843.016

• Transparent reporting form
DOI: https://doi.org/10.7554/eLife.34843.017

### Data availability

All data generated or analysed during this study are included in the manuscript and supporting files.

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
