## [Decision Letter]

Thank you for submitting your article "Mechano-redox control of integrin de-adhesion" for consideration by *eLife*. Your article has been reviewed by three peer reviewers, and the evaluation has been overseen by a Reviewing Editor and Anna Akhmanova as the Senior Editor. The reviewers have opted to remain anonymous.

The reviewers have discussed the reviews with one another and the Reviewing Editor has drafted this decision to help you prepare a revised submission.

Summary:

Passam et al. examine the influence of the oxidoreductase ERp5 on the binding of platelet αIIbβ3 to fibrinogen. They find that the presence of ERp5 decreases platelet adhesion to fibrinogen-coated coverslips under fluid shear stress, and decreases the binding lifetime between αIIbβ3 and fibrinogen in a biomembrane force probe assay. This is attributed to breakage of a key disulfide bond in αIIbβ3 integrin promoted by mechanical stress. The reviewers feel that the paper potentially represents a significant finding of the potential role of force-dependent redox control of integrin activity by ERp5. However, there are a number of conceptual and experimental issues that need to be addressed.

Essential revisions:

1) The absence of experimental data on fibrin is a major limitation. Specifically, the authors argue that reduction of the β3 integrin chain can potentially help drive a switch in binding preference from fibrinogen to fibrin during thrombus formation. However, they present very little data to demonstrate that this is indeed the case. Therefore, the experiments in Figures 1 and 2 that showed ERp5 weakened binding to fibrinogen should be repeated with fibrin.

As it stands, the majority of the data support the parsimonious interpretation that reduction of C177-C184 should decrease integrin affinity for both fibrinogen and fibrin, a result that is in apparent contradiction with ERp5's role in thrombosis. Given this, it seems possible that ERp5 could be acting on additional protein(s) to influence clot formation. These possibilities must be discussed in a revised manuscript.

2) The impact of the C-terminal fragment on platelet "adhesion" is so dramatic as to require additional analysis to assess potential alternative or additional mechanisms beyond cleavage of disulfide bonds. The authors should assess whether the C-terminal fragment directly aggregates platelets. They should also discuss the possibility that in vivo ERp5 might be proteolyzed, allowing the C-terminal domain to activate integrins.

3) The proposed mechanism of strain-induced disulfide cleavage is not supported by the data. Changes in bond lengths and angles cited in the "strain" argument are extremely small relative to the inherent error in crystallographic coordinates, even with the distributions shown. This part of the manuscript can be removed without harm to the essential findings.

4) There are apparent contradictions in the estimates of ERp5 present in platelets, which raises a concern about the physiologic plausibility of their findings. First, this information would be more useful at the beginning of the manuscript. More importantly, the authors conclude that platelets contain 6000-times more ERp5 than αIIβ3. In contrast, a previous study (Zeiler et al., 2014) used quantitative proteomics in murine platelets to determine a copy number of about 100,000 β3 integrin molecules and about 60,000 ERp5 (i.e. PDIA6) copies in platelets. Burkhart et al. (2012) reported that by quantitative MS total human platelet protein is 2 X 107 molecules, corresponding to 1.5 mg of protein/109 platelets; their estimated copy number for αIIbβ3 is about 64,200-83,300 per platelet and their estimated copy number for ERp5 (PDIA6) is 13,300 per platelet. These discrepancies must be discussed in a revised manuscript.

5) A diagram of the authors' proposed model of how de-adhesion from fibrinogen leads to enhanced thrombus formation would be helpful, complete with the authors' estimated concentrations of ERp5 and the percentage of cleaved C177-C184 bonds at different stages.

6) There is a discordance between the percentage of αIIbβ3 undergoing C177-C184 cleavage, which is ·5% over baseline at 2 ·M and ·15% over baseline at "10X" and the reduction in platelet adhesion at 3000 s^-1^, which is ·60%. This should be discussed.

7) The fibrinogen binding studies using the cells expressing wt αIIbβ3 differ from previous reports in the literature in that, as described, no activator (e.g., mAb, PT25-2, DTT, or Mn^2+^) was added. In addition, there was no test of the specificity of the binding (e.g., with a mAb or αIIbβ3 antagonist). Moreover, since only a single data point is shown without an error bar, the authors should indicate the number of experiments performed.

[Editors' note: further revisions were requested prior to acceptance, as described below.]

Thank you for resubmitting your article "Mechano-redox control of integrin de-adhesion" for consideration by *eLife*. Your revised article has been reviewed by three peer reviewers, and the evaluation has been overseen by a Reviewing Editor and Anna Akhmanova as the Senior Editor. The reviewers have opted to remain anonymous.

The reviewers have discussed the reviews with one another and the Reviewing Editor has drafted this decision to help you prepare a revised submission.

Summary:

The reviewers feel that your data showing force-mediated change in redox sensitivity of integrin affinity represents a significant advance in the field. However, there are still serious concerns about your model in which force directly influences the redox sensitivity of the β3 disulfide bond through strain, which is assumed throughout the manuscript.

Essential revisions:

Given the known sensitivity of integrin conformation to force, the data are at least, if not more so, consistent with a model in which force-coupled ligand binding promotes a conformation accessible to the enzyme, without invoking direct force-induced strain; the simulation data do not provide sufficient evidence for the latter. We would be willing to consider a revised manuscript that clearly lays out alternative allosteric models that couple force, conformational change, and sensitivity of the disulfide to reduction to explain your data (strain could be included as part of one model, of course), rather than being founded upon the assumption of force-induced disulfide strain.

[Editors' note: further revisions were requested prior to acceptance, as described below.]

Thank you for resubmitting your work entitled "Mechano-redox control of integrin de-adhesion" for further consideration at *eLife*. Your revised article has been favorably evaluated by Anna Akhmanova (Senior Editor), and a Reviewing Editor.

The manuscript has been improved but there are some remaining issues that need to be addressed before acceptance, as outlined below:

This version has not presented clear alternative models to describe the results. It is clear that the authors, based on the simulation data, believe that strain on the disulfide leads to enhanced reduction, but this model does not consider that strain induced conformational change simply renders it more accessible to ERp5. While it is certainly acceptable to put forward their preferred model, an additional sentence in the first paragraph of the Discussion can concisely deal with the alternative: "Our data suggest that ERp5 cleavage of the disulfide is enabled by ligand- and force-dependent stretching of the sulfur-sulfur bond. However, the experimental data are also consistent with a force-dependent, ligand-bound conformation that provides enhanced access of the disulfide to ERp5. " Unless they can provide a strong rebuttal, an addition along these lines should be added to the Discussion.

---

## [Author Response]

Essential revisions:1) The absence of experimental data on fibrin is a major limitation. Specifically, the authors argue that reduction of the β3 integrin chain can potentially help drive a switch in binding preference from fibrinogen to fibrin during thrombus formation. However, they present very little data to demonstrate that this is indeed the case. Therefore, the experiments in Figures 1 and 2 that showed ERp5 weakened binding to fibrinogen should be repeated with fibrin.

We have now tested this hypothesis and shown it to be incorrect. Platelet adhesion to fibrin under flow is also impaired by ERp5 (Author response image 1).

**Author response image 1. respfig1:** ERp5 triggers αIIbβ3 integrin de-adhesion from fibrin. Platelet adhesion to immobilized fibrin in the absence of presence of 2 µM ERp5 was measured. Fibrin was immobilized in two ways: immobilized fibrinogen was incubated with thrombin and residual thrombin inactivated with hirudin (fbg/IIa), or pre-formed fibrin was directly immobilized (fibrin). Adhesion was measured in the first 4 minutes at a fluid shear rate of 1000 s^-1^. The bars and errors (1 SD) are from three measurements each from three different healthy donor platelets. *, p < 0.05, **, p < 0.01; assessed by unpaired, two-tailed Student’s t-test. Additional experimental details are as described in the manuscript for platelet adhesion to fibrinogen.

As it stands, the majority of the data support the parsimonious interpretation that reduction of C177-C184 should decrease integrin affinity for both fibrinogen and fibrin, a result that is in apparent contradiction with ERp5's role in thrombosis. Given this, it seems possible that ERp5 could be acting on additional protein(s) to influence clot formation. These possibilities must be discussed in a revised manuscript.

The reviewer is correct and we have modified the discussion accordingly. The fourth paragraph in the Discussion now reads:

‘Our studies of ERp5’s role in thrombus formation in vivo (Passam et al., 2015) and ERp5 effects on platelet adhesion under flow shown herein, indicate that ERp5 is involved in propagation of the thrombus and not thrombus initiation. ERp5 accumulates in the developing thrombus (Passam et al., 2015), which may be a mechanism to limit thrombus growth by shifting the balance between fibrinogen cross-linking of platelet αIIbβ3 and fibrinogen dissociation from this receptor. Testing this theory in vivo will be complicated by the likely possibility that ERp5 has more than one substrate in the thrombus. For instance, our preliminary results indicate that ERp5 significantly enhances binding of platelets to von Willebrand factor.’

2) The impact of the C-terminal fragment on platelet "adhesion" is so dramatic as to require additional analysis to assess potential alternative or additional mechanisms beyond cleavage of disulfide bonds. The authors should assess whether the C-terminal fragment directly aggregates platelets. They should also discuss the possibility that in vivo ERp5 might be proteolyzed, allowing the C-terminal domain to activate integrins.

The C-terminal fragment directly aggregates platelets. This result is shown in new Figure 4E (subsection “Both catalytic domains of ERp5 are required for substrate specificity”, last paragraph). Aggregation of washed platelets activated with the PAR-1 agonist, TRAP, in the absence or presence of 2 µM ERp5 or ERp5 CTD was measured. The data points and errors are from 3 different healthy donor platelets.

The possibility that ERp5 is proteolyzed in vivo to expose new functions of the resulting fragments has been discussed in the revised manuscript (Discussion, third paragraph). Although, we have seen no evidence of this to date.

3) The proposed mechanism of strain-induced disulfide cleavage is not supported by the data. Changes in bond lengths and angles cited in the "strain" argument are extremely small relative to the inherent error in crystallographic coordinates, even with the distributions shown. This part of the manuscript can be removed without harm to the essential findings.

This section has been removed from the manuscript.

4) There are apparent contradictions in the estimates of ERp5 present in platelets, which raises a concern about the physiologic plausibility of their findings. First, this information would be more useful at the beginning of the manuscript. More importantly, the authors conclude that platelets contain 6000-times more ERp5 than αIIβ3. In contrast, a previous study (Zeiler et al., 2014) used quantitative proteomics in murine platelets to determine a copy number of about 100,000 β3 integrin molecules and about 60,000 ERp5 (i.e. PDIA6) copies in platelets. Burkhart et al., 2012 reported that by quantitative MS total human platelet protein is 2 X 107 molecules, corresponding to 1.5 mg of protein/109 platelets; their estimated copy number for αIIbβ3 is about 64,200-83,300 per platelet and their estimated copy number for ERp5 (PDIA6) is 13,300 per platelet. These discrepancies must be discussed in a revised manuscript.

Thank you for pointing out these studies, which have been discussed in the Introduction of the revised manuscript (fifth paragraph). We have now tested the veracity of our ELISA results using an immunoblotting approach, with reference to a standard curve generated using recombinant ERp5, to estimate the level of platelet ERp5 and quantity released into the supernatant upon activation (new Figure 1—figure supplement 1). We estimate that a human platelet contains ~47,500 molecules of ERp5 and release approximately half of this into the supernatant upon activation (subsection “ERp5 triggers fibrinogen dissociation from αIIbβ3 integrin”, first paragraph). This ERp5 level is in the range of the literature values. Our ELISA was reporting spuriously high values and has been removed from the revised manuscript.

5) A diagram of the authors' proposed model of how de-adhesion from fibrinogen leads to enhanced thrombus formation would be helpful, complete with the authors' estimated concentrations of ERp5 and the percentage of cleaved C177-C184 bonds at different stages.

A diagram of our proposed model appears in new Figure 7 of the revised manuscript. Inclusions of estimated ERp5 concentrations and fraction of cleavage of the C177-C184 bond would be overstating the experimental evidence we have at this stage.

6) There is a discordance between the percentage of αIIbβ3 undergoing C177-C184 cleavage, which is ·5% over baseline at 2 ·M and ·15% over baseline at "10X" and the reduction in platelet adhesion at 3000 s^-1^, which is ·60%. This should be discussed.

This is a good point. The following discussion has been added to the revised manuscript (Results, subsection “ERp5 cleaves the βI-domain Cys177-Cys184 disulfide bond): ‘Approximately 30% of the Cys177-Cys184 disulfide bond in the purified integrin preparation is cleaved by 10-fold molar excess of ERp5 under static conditions, which is a ~20% increase over baseline. […] ERp5-mediated dissociation of fibrinogen from αIIbβ3 is greatly enhanced when force goes beyond 15 pN and is complete at 40 pN (Figure 2D).’

7) The fibrinogen binding studies using the cells expressing wt αIIbβ3 differ from previous reports in the literature in that, as described, no activator (e.g., mAb, PT25-2, DTT, or Mn^2+^) was added. In addition, there was no test of the specificity of the binding (e.g., with a mAb or αIIbβ3 antagonist). Moreover, since only a single data point is shown without an error bar, the authors should indicate the number of experiments performed.

This result has been expanded in the revised manuscript (Results, subsection “Cleavage of the βI disulfide results in reduced affinity for fibrinogen due to increased βI-domain flexibility and high stresses at the MIDAS site”, first paragraph). Binding of soluble or immobilized fibrinogen to HEK cells expressing wild-type or disulfide mutant (C177,184S) αIIbβ3 integrin has been measured in the absence or presence of the integrin activator, Mn^2+^ (new Figure 5). Expression of wild-type (64.8% of cells) and disulfide mutant integrin (82.9% of cells) was confirmed by staining with anti-β3 antibody (new Figure 5A). Soluble fibrinogen bound to less than 20% of wild-type or disulfide mutant β3 positive cells in the absence of Mn^2+^ (new Figure 5B). In the presence of Mn^2+^, binding of soluble fibrinogen to αIIbβ3 with a broken Cys177-Cys184 disulfide bound was impaired (p < 0.05) compared to wild-type integrin (new Figure 5B). Cells expressing wild-type integrin adhered to immobilized fibrinogen in the absence and presence of Mn^2+^, whereas adherence of the disulfide mutant integrin was severely impaired in both conditions (p < 0.0001 in the absence of Mn^2+^ and p < 0.001 in the presence of Mn^2+^) (new Figure 5C). Data points and errors (SD) are from 3-4 biological replicates.

[Editors' note: further revisions were requested prior to acceptance, as described below.]

Essential revisions:Given the known sensitivity of integrin conformation to force, the data are at least, if not more so, consistent with a model in which force-coupled ligand binding promotes a conformation accessible to the enzyme, without invoking direct force-induced strain; the simulation data do not provide sufficient evidence for the latter. We would be willing to consider a revised manuscript that clearly lays out alternative allosteric models that couple force, conformational change, and sensitivity of the disulfide to reduction to explain your data (strain could be included as part of one model, of course), rather than being founded upon the assumption of force-induced disulfide strain.

Thank you for the helpful comments. We agree that our results are best described by a force-coupled ligand binding mechanism. We had poorly conveyed this message in the manuscript.

The text of the manuscript has been revised to clearly describe this mechanism. The Abstract, Introduction (last paragraph), Discussion (first paragraph) and Figure 7 have been revised.

[Editors' note: further revisions were requested prior to acceptance, as described below.]

The manuscript has been improved but there are some remaining issues that need to be addressed before acceptance, as outlined below:This version has not presented clear alternative models to describe the results. It is clear that the authors, based on the simulation data, believe that strain on the disulfide leads to enhanced reduction, but this model does not consider that strain induced conformational change simply renders it more accessible to ERp5. While it is certainly acceptable to put forward their preferred model, an additional sentence in the first paragraph of the Discussion can concisely deal with the alternative: "Our data suggest that ERp5 cleavage of the disulfide is enabled by ligand- and force-dependent stretching of the sulfur-sulfur bond. However, the experimental data are also consistent with a force-dependent, ligand-bound conformation that provides enhanced access of the disulfide to ERp5." Unless they can provide a strong rebuttal, an addition along these lines should be added to the Discussion.

The suggested change has been made to the manuscript (Discussion, first paragraph).